# Multi-Modal Prehabilitation in Thoracic Surgery: From Basic Concepts to Practical Modalities

**DOI:** 10.3390/jcm13102765

**Published:** 2024-05-08

**Authors:** Marc Licker, Diae El Manser, Eline Bonnardel, Sylvain Massias, Islem Mohamed Soualhi, Charlotte Saint-Leger, Adrien Koeltz

**Affiliations:** 1Department of Cardiovascular & Thoracic Anaesthesia and Critical Care, University Hospital of Martinique, F-97200 Fort-de-France, France; diae.elmanser@chu-martinique.fr (D.E.M.); eline.bonnardel@chu-martinique.fr (E.B.); sylvain.massias@chu-martinique.fr (S.M.); soualhi.islem@gmail.com (I.M.S.); adrien.koeltz@chu-martinique.fr (A.K.); 2Faculty of Medicine, University of Geneva, 1206 Geneva, Switzerland; 3Department of Cardiovascular & Thoracic Surgery, University Hospital of Martinique, F-97200 Fort-de-France, France; charlotte.renard@chu-marinique.fr

**Keywords:** surgical stress, frailty, aerobic capacity, exercise training, nutrition, patient empowerment

## Abstract

Over the last two decades, the invasiveness of thoracic surgery has decreased along with technological advances and better diagnostic tools, whereas the patient’s comorbidities and frailty patterns have increased, as well as the number of early cancer stages that could benefit from curative resection. Poor aerobic fitness, nutritional defects, sarcopenia and “toxic” behaviors such as sedentary behavior, smoking and alcohol consumption are modifiable risk factors for major postoperative complications. The process of enhancing patients’ physiological reserve in anticipation for surgery is referred to as prehabilitation. Components of prehabilitation programs include optimization of medical treatment, prescription of structured exercise program, correction of nutritional deficits and patient’s education to adopt healthier behaviors. All patients may benefit from prehabilitation, which is part of the enhanced recovery after surgery (ERAS) programs. Faster functional recovery is expected in low-risk patients, whereas better clinical outcome and shorter hospital stay have been demonstrated in higher risk and physically unfit patients.

## 1. Introduction

Thoracic surgery encompasses a large spectrum of diagnostic, curative and palliative procedures for diseases affecting the lungs, the airways, the esophagus, the chest wall, the mediastinum, and the diaphragm [1].

In the early days of thoracic surgery, pulmonary tuberculosis and other infections causing lung abscesses were frequent indications for intra-thoracic interventions (e.g., chest tube insertion, thoracoplasty, lung resection), some of them being now considered more deleterious than beneficial (e.g., artificial pneumothorax to collapse the infected lung). With the advent of antibiotics after 1943, the role of thoracic surgery has shifted to cancer treatment and removal of infectious foci. Given silent progression of cancer in the lungs, the pleura and the esophagus, less than 30% of patients are suitable candidates to undergo curative treatment. Nowadays, with better diagnostic tools, a growing number of elderly patients are diagnosed with early stages of thoracic cancer. However, in some patients, the burden of chronic illnesses and the severity of organ dysfunction may contraindicate surgery when the risks of death or debilitating complications overwhelm the benefits of a curative resection. Indeed, poor tolerance to surgical stress owing to comorbidities and a frail condition paves the way to postoperative pulmonary and cardiovascular complications (PPC and PCVC, respectively) in as much as 20% to 50% of patients [2]. These PPCs and PCVCs lead to frequent admission in the intensive care unit (ICU), prolonged hospital length of stay and poor survival [3].

Over the past three decades, major technological advances have fostered the development of minimally invasive procedures with enhanced workability and vision allowing more precise surgical manipulation that have resulted in lesser tissue damage and greater safety [4]. Meanwhile, anesthesiologists manage the upper airways with new devices (double-lumen tubes, bronchial blockers) and under endoscopic guidance [5]. Better understanding of the mechanisms of postoperative complications has been associated with implementation of lung protective strategies, optimization of hemodynamic status and provision of a multimodal analgesic regimen [6].

In the early 1990s, the concept of “fast track” surgery was introduced by a Danish surgeon, Henri Kehlet, who emphasized the importance of basic perioperative principles, such as shortening the fasting period, performing smaller incisions, maintenance of body homeostasis (e.g., fluids, temperature, glycemia), as well as early mobilization, feeding and removal of all catheters and drains shortly after surgery [7]. Nowadays, this concept has evolved towards an integrative clinical care pathway for different types of surgical procedures and has been coined the *Enhanced Recovery after Surgery* (ERAS) program [8]. Health care workers (e.g., surgeons, anesthesiologists, oncologists, physical therapists, nurses) collectively are asked to describe all processes of care in the pre-intra and postoperative periods [9]. This team-based multimodality approach aims to optimize patient condition, reduce surgical stress response, and facilitate postoperative recovery. To standardize all care processes, the ERAS team selects a bundle of interventions based on physiological rationales, published clinical evidence and collective experience. Within the ERAS program, all interventions aiming to enhance physiological patient’s reserve and reinforce patient’s tolerance to sustain surgery are grouped under the concept of “prehabilitation” and entail three domains: (1) optimization of medical treatment and nutritional support, (2) patient’s education for healthy behavior and (3) exercise training [9]. In contrast to prehabilitation, the term “rehabilitation” refers to similar interventions conducted in two different populations of patients, those presenting with chronic debilitating diseases (e.g., chronic obstructive pulmonary disease, heart failure, stroke, diabetes mellitus) and those recovering from surgery and presenting with functional deficits [10,11,12].

In this review paper, we will first describe the homeostatic body responses to surgery, then highlight the preoperative risk factors of postoperative complications, describe the functional assessment, and finally address the components of prehabilitation in the clinical pathway of patients undergoing thoracic surgery.

## 2. Surgical Stress and Physiological Responses

### 2.1. Neuroendocrine and Inflammatory Pathways

Surgical tissue trauma and organ manipulation trigger a variable stress response characterized by stimulation of the hypothalamic–pituitary–adrenal (HPA) axis and sympathetic nervous system (SNS), along with activation of the renin–angiotensin aldosterone system (RAAS) [1,13]. Accordingly, the surgical-induced release of adrenocorticotropic hormone (ACTH), cortisol, catecholamines, aldosterone, arginine vasopressin (AVP), growth hormone and glucagon attempts to provide sufficient energy substrates to fuel the healing processes and enhance oxygen delivery while maintaining cardiovascular homeostasis (Figure 1).

Besides this neuroendocrine response, inflammatory mediators induced by tissular damage are collectively termed “damage associated molecular patterns” (DAMPs) or alarmins, whereas those induced by eventual later infection are called “pathogen associated molecular patterns” (PAMPs) [14]. In addition to producing acute inflammation, both signals affect the innate immunity via the Toll like receptor 4 (TLR4) signaling pathway and the capacity to eliminate (or tolerate) various microorganisms and foreign bodies [15].

Local inflammation is accompanied by a systemic inflammatory response syndrome (SIRS), which is proportional to the severity of the initiating traumatic insult and is determined by the balance between pro- and anti-inflammatory mediators [13,16].

Cytokines (e.g., interleukins (IL), chemokines, interferons and tumor necrosis factors) (TNF) are directly produced at the surgical site by macrophages, neutrophils, dendritic cells and non-killer (NK) cells [17]. Proinflammatory mediators (e.g., TNF-alpha and interleukins (IL) such as IL-6, IL-1 beta, and IL-8) cause transient fever with the production of acute phase proteins (APP) in the liver (e.g., fibrinogen, C-reactive protein, D-dimer, alpha2-marcoglobulin). Along with the release of DAMPs and PAMPs, immunosuppression results from the predominant anti-inflammatory effects of specific cytokines (e.g., IL-4, IL-10, IL-1 receptor antagonist), and the shift in Thelper-1:Thelper-2 cells (Th-1, Tth-2) owing to inhibition of Th-1 cells. The magnitude of these processes is associated with a propensity to develop sepsis and later cancer recurrence.

### 2.2. Biological and Clinical Expression of the Stress Response

The mild and transient increases in body temperature (37–38 degree Celsius) and in respiratory rate reflect the hypermetabolic state induced by inflammatory mediators [18]. In healthy individuals, the increase in systemic oxygen consumption (+5 to 50% elevation) is matched by increases in oxygen transport and tissue extraction capacity as expressed by a mild tachycardia and enhanced cardiac output in response to catecholamine released from the activated SNS and adrenal medulla. Given the hypothalamic release of AVP in response to SNS and RAAS stimulation, fluid retention and relative oliguria are common in the early postoperative days. Although these mechanisms are helpful to maintain circulatory volume and cardiac preload, there are also incriminated in body weight gain and poor wound healing due to excessive extracellular water accumulation. At least two of the following criteria are required to qualify for SIRS: central temperature > 38 °C or <36 °C, heart rate > 90 beats/min, respiratory rate > 20 breaths/min (or partial carbon dioxide pressure < 32 mmHg), white blood cell count > 12 × 10^9^/L or <4 × 10^9^/L [16]. Given the neuroendocrine and inflammatory stimulation, the SIRS is associated with enhanced lipolysis, glycogenolysis and insulin-resistance as well as degradation of myofibrillas from skeletal muscles in the immobilized surgical patients.

The catabolic pathway tends to maximize the delivery of energetic substrates from glycogen and fat, whereas the anabolic pathway is characterized by diversion of amino acids to produce APP in the liver, resulting in muscle wasting that is aggravated in cancer patients, and by septic postoperative complications [19].

Sarcopenia often coexists with preexisting anemia, neural dysautonomia, impaired ventricular function and increased vascular stiffness [20]. Systemic inflammatory processes involving the release of reactive oxygen species (ROS) and cytokines, such as TNF-α and IL-1, have been implicated in deregulation of mitochondrial function and degradation of striated muscle proteins through all four proteolytic systems (i.e., calpains, caspase-3, ubiquitin-proteasome system and autophagy) [21]. Interestingly, among all skeletal muscles, the diaphragm is most prone to inactivity- and inflammatory-induced accelerated proteolysis and decreased protein synthesis, as well as mitochondrial oxidative stress and disruption of calcium homeostasis [22]. Hence, the combination of surgery-induced inflammation and mechanical unloading of respiratory muscles triggers the degradation of the myofibrillar component, leading to muscle atrophy and poor contractile performance. These mechanisms explain the development of atelectasis and difficulties in weaning from the ventilator when the patient awakens from prolonged anesthesia and faces increased ventilatory workload due to incisional pain and interstitial lung edema [23].

## 3. Postoperative Complications

Given the lack of objective criteria used to define adverse events, the incidence of complications after thoracic surgery varies within a large range (10–80%) [24]. In 2015, a European joint taskforce published guidelines to define perioperative clinical outcome (EPCO) based on objective criteria [25]. The proposed classification system discriminates between physiological derangements induced by surgery (e.g., fatigue, need for increased fraction of inspired oxygen (FIO2)), variable levels of organ dysfunction and quality of life impairments (e.g., EQ-5D, SF-6, WHO disability assessment schedule). Ultimately, a panel of experts in perioperative medicine has recommended assessment of the severity of any PC based on the need for medical or surgical treatment and the use of composite outcome measures by grouping individual adverse events [24]. Such EPCO approach allows a more precise identification of risk factors, along with better assessment of risk-minimizing intervention among the surgical population. Based on these guidelines and an expert-based consensus, we propose an integrative hierarchical model to appraise the perspectives of different stakeholders and to include objective measures, such as organ dysfunction of increasing severity and patient-centered values (Table 1) [26].

After thoracic surgery, PPCs, i.e., atelectasis, pneumonia, acute respiratory distress syndrome or acute lung injury (ARDS or ALI), broncho-pulmonary fistula and pleural effusions, outnumber PCVCs (atrial fibrillation, heart failure and myocardial infarct). Pneumonia occurs in 5 to 15%, particularly in patients with deficient immune status, pre-existing airway colonization/contamination, bronchoalveolar collapse or broncho-aspiration [27,28,29]. More recently, myocardial damage has been diagnosed by serial measurements of cardiac troponin in up to 27% of patients undergoing thoracic surgery, even in the absence of clinical expression of PCVC [30]. This so called myocardial injury after surgery (MINS) is a strong predictor of early and mid-term survival [31].

Taken together, 30-day mortality is largely determined by the extent of surgical resection, bleeding and ARDS/ALI, often associated with pneumonia or other septic condition [32,33]. In contrast, long-term outcome following lung cancer resection is predominantly influenced by the occurrence of early PPCs and MINS, as well as by respiratory failure and recurrence of cancer [34,35].

## 4. Preoperative Assessment and Risk Factors for Postoperative Complications

### 4.1. Clinical Assessment

At the preoperative visit, the anesthesiologist plays a crucial role, acting as a “gatekeeper” by judging the patient’s ability to sustain the surgical procedure and by mitigating the stress response with a proper anesthetic strategy, while optimizing medical treatments and enhancing physiologic reserves before surgery [36].

The occurrence of an adverse event is determined by the interactions between three main components: (1) patient’s risk factors (physiological reserves, psychological condition and social environment), (2) the burden of surgical trauma, (3) the individual and collective skills of health care professionals, as well as the logistic aspects in the clinical care processes (e.g., availability of intensive care beds, rescue teams) [37].

In non-cardiac surgery, preoperative risk assessment is mainly based on medical history and clinical evaluation using the American Society of Anesthesiology Physical Status (ASA-PS) score, the Revised Cardiac Risk Index (RCRI) and the Assess Respiratory Risk in Surgical Patients in Catalonia (ARISCAT) score (Table 2) [38]. Nutritional status, smoking habit, alcohol consumption and corticoid treatment are additional risk factors for postoperative complications.

### 4.2. Cardiac and Respiratory Assessment

Patients with unstable cardiac condition (e.g., recent myocardial infarct, sustained arrhythmias, syncopal event), heart murmur with dyspnea, RCRI > 2 and/or poor exercise tolerance should be sent to a cardiologist for further investigations. Some of these patients may benefit from myocardial revascularisation or adjustment of drug treatment.

Besides chest imaging, all thoracic patients undergo measurements of lung volumes, gas flow and diffusion capacity of carbon oxide (DLCO) to document chronic obstructive pulmonary disease and restrictive lung disease. Considering the extent of lung resection, the postoperative functional condition can be calculated from preoperative forced expiratory volume over the first second (FEV_1_) and DLCO. Predicted postoperative FEV1 and DLCO less than 40% and/or hypercapnia (PaCO_2_ > 7.5 kPa) at rest are considered contraindications for lung resection and these patients should undergo further medical optimization or alternative non-surgical treatments.

### 4.3. Functional Assessment

The functional capacity can simply be addressed by questionnaires to evaluate exercise tolerance (Metabolic Equivalent Task, (MET)) and daily life activities (Duke Activity Status Index (DASI)), or by dynamic physical tests (e.g., time up to go, gait speed) [39,40,41]. More recently, the clinical frailty scale based on the need for assistance in daily life activities has been shown as useful to complement risk stratification in the most vulnerable patients [42].

Before thoracic surgery, cardiopulmonary exercise testing (CPET) on a cycloergometer or a treadmill represents the reference tool to quantitate aerobic fitness by measuring peak and maximal oxygen consumption (peakVO_2,_ maximalVO_2_), anaerobic threshold, peak workload and ventilatory efficiency (slope or ratio of ventilation to carbon dioxide production) [43]. The CPET-derived measurements reflect the integrative response of the respiratory, circulatory and muscular systems during maximal exercise [44]. Alternatively, low technology exercise tests (e.g., shuttle, stair climbing, six-minute walk distance) can be used as screening tools or when CPET is not available [45].

Poor aerobic physical fitness is primarily dependent on ventilatory impairments (respiratory muscle and gas exchange capacity), insufficient oxygen transport (cardiac and vascular components, hemoglobin level) and/or skeletal muscle limitations (muscular deconditioning, joint disorders or neurological deficits) [46,47]. Low aerobic fitness (less than 12-15 mL/kg peak VO_2_) is reported in up to 20–30% patients scheduled for lung cancer surgery and is predictive of poor survival [48]. Likewise, sedentary individuals and patients with chronic inflammatory diseases, coronary artery disease (CAD), heart failure (HF), chronic obstructive pulmonary disease (COPD) and neurological disorders are all characterized by an impaired cardiopulmonary exercise tolerance and a reduction in lean body mass that both represent risk factors for diminished long term survival [49].

## 5. Implementation of a Prehabilitation Program

The term “prehabilitation” was coined in 1942 by US military physicians as a means to remediate the poor physical condition detected in more than 50% of soldiers enlisted for medical examination [50]. The combination of physical training, as well as good housing, diet and hygiene, was found to improve the health rating of 85% of 12,000 men who participated in a study in 1946 [51]. More than 40 years elapsed until the concept was adopted in sport medicine to prevent injuries and in clinical medicine for various chronic and acute illnesses [52]. In 2002, Topp et al. proposed a generic program of prehabilitation including aerobic training, strength exercises and functional and flexibility components aiming to counteract muscular wasting resulting from bedrest and inflammation in critically-ill patients admitted to the intensive care unit [53].

In the modern era, prehabilitation programs highlight the key role of patients at the core of all healthcare decisions, in order to comply with treatment interventions in the perioperative journey [54]. By providing holistic, person-centered, individualized pre-operative optimization strategies, patients and families feel empowered, motivated, and in control of their own health. A coordinating nurse plans all preoperative consultations, examinations and therapeutic sessions while being the referent person for the patient and her/his family (Figure 2) [55]. The American Society of Clinical Oncology recommends preoperative exercise and specific diet protocols for patients with lung cancer undergoing surgery to improve their outcomes and wellbeing [56].

All patients may benefit from prehabilitation, which is part of the ERAS program. Faster functional recovery is expected in low risk and fit patients, whereas better clinical outcome and shorter hospital stay have been demonstrated in higher risk and physically unfit patients (Figure 3).

### 5.1. Optimized Medical Treatment and Correction of Nutritional Deficits

At the preoperative visit, the anesthesiologist ensures that the patient’s chronic illnesses are properly managed according to updated professional guidelines, particularly heart failure, coronary artery disease and chronic kidney disease.

In case of a new or worsening inflammatory state, any infection should be ruled out and treated with antibiotics before elective surgery. Attention should also be paid to prescribing continuation or/withdrawal of medications that influence cardiovascular homeostasis and the risk of bleeding/thrombosis.

Laboratory investigations detect anemia (hemoglobin < 12 g/dL in women and <13.5 g/dL in man), poorly controlled glycemia (hemoglobin A1c (HbA1c) > 7.5%) and nutritional defects (low serum levels of prealbumin, leptine and vitamin D) [57]. Preoperatively, anemia is prevalent in as much as 20% to 40% of lung cancer patients and supplementation with iron, folic acid and vitamine B12 has been shown as effective in reducing the need for transfusion and the occurrence of postoperative complications [58,59,60]. In a meta-analysis of 8 RCTs including 1450 surgical patients with anemia, preoperative treatment with erythropoietin was associated with a lesser need for allogenic transfusion (RR 0.83, 95% confidence interval (CI) 0.69–0.99, *p* = 0.049) and a statistically non-significant increase in thrombotic events (RR 1.32 (95%CI = 0.88–1.972, *p* = 0.180) [61].

Nutrition screening tools, such as the Mini-Nutritional Assessment (MNA), are useful in detecting malnutrition, which has been reported in 10–50% of surgical candidates and is associated with lower 5-year postoperative survival [62]. The causes of malnutrition are multifactorial, being related to lung diseases (chronic obstructive pulmonary disease, cancer-induced inflammation with loss of appetite, recent pneumonia), other comorbid conditions (e.g., heart failure, gastrointestinal or liver disease), medications, physical disability (e.g., poor dentures), or socio-economic factors (e.g., low income) [63]. Depending on the underlying cause, preoperative correction of nutritional deficit may not be possible and, when associated with low aerobic capacity, dependent status and/or sarcopenia, it may represent a contraindication to surgery. Undernourished patients may benefit from personalized diets over 4 to 12 weeks to replenish muscle mass while restoring muscular strength and aerobic fitness [63]. Dietary adjustments are preferentially made by prescribing the intake of high energy nutrients (~30–40 kcal/kg/day, carbohydrates, omega-3 fatty acids), high-quality source of proteins (~1.5–2 g/kg/day of protein, creatine monohydrate, essential aminoacids) and selective supplements of vitamins and trace elements (e.g., vitamin D, folic acid, cyanocobalamin, iron) [64]. Provision of these multi-ingredient mixtures in the elderly has demonstrated favorable effects on lean body mass and muscular strength, with further gains when nutrition support was combined with resistance and aerobic exercise training [65]. Postoperatively, attention should be paid to resume enriched feeding in these frail patients, preferentially orally or, if not possible, parenterally.

### 5.2. Patient Education and Hygienic Interventions

Several lifestyle behaviors have been associated with reduced occurrence of postoperative complications and better patient’s wellbeing [66,67]. Empowerment education refers to the provision of disease-related knowledge and disease management, enabling patients to face up their diseases, make behavior changes and voluntarily engage in the prehabilitation program [68].

Tobacco and alcohol dependency are frequently reported among patients with lung cancer (up to 80% and 25%, respectively). Chronic exposure to these toxic agents is associated with impaired tissue healing, poor immune response, along with increased risk of infections and cancer recurrence [69,70] Therefore, smoking cessation is the most important lifestyle change to maximize the benefits of curative surgery and chemotherapy while improving patient’s quality of life. Smoking should ideally be stopped at least 3 weeks before surgery. [71]. Behavioral support and pharmacological interventions (nicotine substitution, varenicline, and bupropion) are effective strategies for smoking cessation, which can be achieved in 30 to 50% of patients (compared with less than 25% among non-surgical patients) [72]. In contrast, few data support the use of electronic nicotine device systems in perioperative smoking cessation [73,74]. Preoperative alcohol abstinence (4 to 8 weeks) has been associated with improved immune status, lesser bleeding and fewer postoperative arrhythmias [75]. Alcohol-induced sarcopenia and alterations of the liver, brain, pancreas, heart and autonomic nervous system require longer periods of abstinence and supplemental nutritional support. Withdrawal syndrome should be anticipated and consultation with an addiction specialist should be considered.

Prevention of surgical site infection (SSI) encompasses the eradication of any preexisting distant infectious foci (hematogenous spread), as well as systematic decolonization of the skin and nose by repeated antiseptic showers [76,77]. Odontogenic foci and periodontitis frequently develop in patients with cancer, particularly those receiving chemotherapy, and heavy drinkers. The dental plaque includes a biofilm colonized with Gram-negative bacteria that can be transferred to the respiratory tract and pulmonary alveola. Mouth rinsing and toothbrushing with chlorhexidine solution has been shown effective in lowering the risk of postoperative pneumonia [78]. In a Japanese nationwide database involving major cancer surgery (N = 509,179), preoperative oral care by a dentist was associated with lower rates of postoperative pneumonia and 30-day mortality [79]. Finally, clearing Staphylococcus aureus and Gram-negative bacteria from the nose and skin can be achieved by using mupirocin nasal ointment twice daily and showering the body with skin antiseptic (chlorhexidine or triclosan preparation) for 3–5 days before surgery [76,80].

### 5.3. Exercise Training

#### 5.3.1. Type of Exercise

Physical activity falls into two main categories, dynamic and static exercises, i.e., endurance and resistive training (ET and RS), that are complemented with stretching, balance and flexibility exercises [81]. Building up muscular mass is usually achieved by “resistive work” and strengthening exercises with isometric contractions and high mechanical loads [82]. In contrast, improvement in aerobic capacity following ET results from major increases in gas exchange and cardiac output that are associated with repeated concentric and/or eccentric muscular contractions [83]. Short sessions of high intensity interval training (HIIT) sessions compared with moderate intensity ET have been shown effective to maximize aerobic capacity within a relatively short period, even in the elderly [84]. Many physical activities fit into more than one category, ET also contributes to enhance lean body mass, whereas resistive work may promote body balance along with cardiac hypertrophy [85]. Interestingly, respiratory muscle training (RMT) using volume incentive spirometry or resistive threshold loading devices has been shown effective in boosting physical performances in healthy subjects [86] and improving daily life autonomy of patients with chronic disabilities [87]. This is particularly valuable in patients with neuromuscular or joint disabilities and those at risk of myocardial infarct (e.g., severe coronary artery disease) and sudden death (e.g., critical aortic stenosis) where high peakVO_2_ cannot be achieved or is considered too risky [88,89]. Finally, prescription of concurrent aerobic and resistive training in the elderly has been shown effective in improving both functional capacity and muscle performances [90].

#### 5.3.2. Mechanisms of Training-Induced Improvements

In skeletal muscles, nuclear factors, such as the peroxisome proliferator-activated receptor (PPAR) and their co-regulators, sirtuin (SIRT) and adenosine monophosphate activated protein kinase (AMPK), play important roles in sensing energy homeostasis, coordinating metabolic flux and upregulation of genes involved in fatty acids and glucose uptake and oxidation [91]. ET induces PPARβ/δ overexpression in skeletal muscle resulting in muscular hyperplasia (type I, slow-twitch fibers), angiogenic response and a shift from type II fast-twitch fibers I towards oxidative type I muscle fibers [92]. In contrast, resistive exercise promotes muscle hypertrophy (type II, fast-twitch fibers) through the enhanced expression of Insulin Growth Factor (IGF-1) and may prevent muscle atrophy via an Akt- and Foxo-1-dependent signaling pathway coupled with downregulation of MuRF-1 and pro-inflammatory mediators, such as TNF-α, IL-1β and IL-6 (Figure 3) [93,94].

The ET-induced increase in VO_2_Max involves partial reversal of endothelial dysfunction with higher capillary density, expansion of the blood volume, increased adrenergic receptor responsiveness, improved ventricular relaxation, restoration of insulin sensitivity and enhanced oxidative mitochondrial performances in skeletal muscles, owing to upregulation of PPAR and key enzymes in the tricarboxylic acid cycle [95]. Accordingly, the enhanced cardiac output facilitates tissue oxygen diffusion and the mitochondrial biogenic changes lead to better utilization of oxygen within the working muscles. Both in RMT and ET, the higher ventilatory loads result in structural and functional adaptive changes within respiratory muscles, conferring greater strength and resistance to fatiguing contraction while reducing the metaboreflex [96].

#### 5.3.3. Clinical Medicine

Sedentary individuals and elderly are characterized by reduced expression of PPARγ coactivator-1-α and of mitochondrial activity in skeletal muscles. Since physical training upregulates the expression of PPARγ coactivator-1-α and increases the protein content of the electron transport chain complexes in mitochondria, ET represents an effective intervention to counteract the effects of aging and chronic diseases on mitochondrial biogenesis, oxidative capacity and muscle mass development [97,98]. Likewise, resistance training at moderate loads induces hypertrophic changes of type II fibers with increased muscle strength, these effects being augmented by the intake of dietary components (e.g., proteins, macronutrients) and nutritional supplements (e.g., creatine, vitamin-D, omega-3 polyunsaturated fatty acids) [99,100]. In a meta-analysis of seven trials including 248 older individuals, inspiratory muscle performance was significantly improved after IMT at moderate intensity levels (30–80% of maximal inspiratory pressure) over at least 4 weeks compared with sham treatment [101].

In master endurance athletes (older than 60 yrs), chronic endurance training (ET) results in lesser decline in muscle strength and in higher aerobic capacity compared with age-matched controls (~43 mL/kg/min vs. 27 mL/kg/min VO_2_Max) [34].

In individuals with and without cardiovascular diseases, the practice of regular physical activities has been shown to confer cardioprotective benefits, to decrease the risk of cancer (breast, gastric, liver, colon, and lung), to preserve cognitive function and to prolong life free from severe disabilities [102,103]. In patients with COPD, heart failure and various neuromuscular disorders, rehabilitation programs conducted over 6 to 12 weeks have been shown effective in improving exercise tolerance and quality of life [10,11,12].

#### 5.3.4. Physical Training Program before Major Surgery

Compared with healthy individuals, surgical patients with cancer, COPD or cardiovascular disease exhibit an average 20–40% reduction in aerobic capacity and inspiratory muscle strength, making them more vulnerable to postoperative complications, particularly PPC [104]. Therefore, hospital- or home-based training modalities need to be tailored to the short preoperative time frame (1–3 weeks), as well as to individual’s limitations and preferences. Most patients are capable of increasing their aerobic fitness by 1.6 to 2 mL/kg/min and maximal inspiratory pressure by an average of 15 cm of water (+18%) which are considered clinically and functionally relevant changes [105,106]. The training-induced physiological improvements are inversely related to baseline fitness level and directly related to the training load as expressed by the cumulative sum of the product of exercise intensity and duration of each training session [107].

A meta-analysis including 29 RCTs undergoing cardiac, lung and major abdominal surgery (N = 2070 patients) strongly supports the effectiveness and safety of ET, RMT or a combination of both to reduce the occurrence of PPCs and to shorten the hospital length of stay [106]. Reversal of respiratory muscle weakness and increased aerobic capacity was achieved even after one week when an intensive training program was prescribed. These ET-induced protective mechanisms involve cardiovascular and muscular adaptive changes (e.g., enhanced oxygen tissue delivery and extraction), allowing the postoperative patient to sustain higher ventilatory loads and to prevent alveolar collapse while facilitating clearance of bronchial secretions.

In the subset of patients undergoing lung cancer resection, although preoperative exercise training failed to achieve significant decrease in short-term mortality (RR = 0.66, 95% CI 0.22 to 2.22), the overall incidence of major complications based on the Dindo–Clavien score ≥ 2 was reduced (RR = 0.42, 95% CI 0.25 to 0.69), while indicators of quality of life tended to improve.

A rapid increase in cardiorespiratory fitness can be elicited with HIIT, which involves repeated bursts of physical work to achieve approximately 80% of the maximum heart rate (30–60 s followed by 60–90 s recovery) [108,109]. Such a preoperative training program is appealing, in order to trigger protective cellular pathways and mitochondrial biogenesis even in elderly and patients with comorbidities [110]. A meta-analysis of 12 RCTs with patients undergoing major surgery (N = 772) suggested that HIIT led to enhanced aerobic capacity (+2.6 mL/kg/min peakVO_2_) and was associated with fewer postoperative complications (−53%) [111].

## 6. Conclusions

Implementation of an effective prehabilitation protocol represents a truly multidisciplinary endeavour where anaesthetists, surgeons, oncologists, pneumologists, physiotherapists, specialist nurses and dieticians all have important roles to play, cooperating with each others and interacting with the patient.

Patients with lung cancer may benefit from prehabilitation within the ERAS program by improving their functional recovery and decreasing the incidence of adverse events, along with lower medical costs and shorter hospital length of stay.

There are synergistic effects between the three pillars of prehabilitation: optimizing patients’ comorbid and nutritional condition, improving exercise tolerance, and moving towards a healthier lifestyle through education and psychological support, which should be continued after discharge from the hospital (Figure 4).

There is a sound physiological rationale and growing scientific evidence for training-induced improvement in aerobic capacity and for nutrition-induced increase in lean body mass within 1 to 4 weeks before surgery with the aim of enhancing the patient’s ability to sustain the surgical stress. Continuation of the exercise training program and adhesion to a healthier lifestyle are necessary to consolidate functional gains and increase patient’s life expectancy. Besides hospital and cancer-based prehabilitation protocols, home-based virtual interventions (voluntary attendance at Zoom mind–body fitness classes) has been shown effective in delivering preoperative education in remote areas and in lowering hospital readmission after cancer resection s [112].

Future studies will help to design an individualized optimization approach based on a better understanding of the complex interplay between the patient’s genetic background, pathophysiological responses to surgery and social environments.

## Figures and Tables

**Figure 1 jcm-13-02765-f001:**
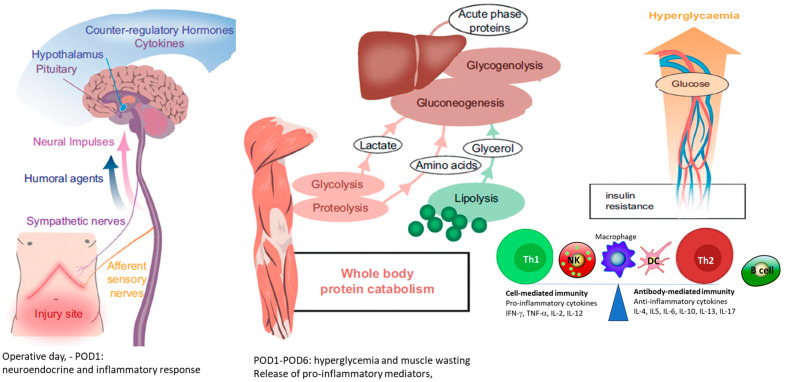
Time course of neuroendocrine and inflammatory response to surgery. DC, dendritic cell; IL, interleukin; IFN-γ, interferon gamma; TGF-β, transforming growth factor-β. Th2 and Th1, T helper 1 and 2 cells; POD1,3, and x, postoperative day 1, 3 and undetermined.

**Figure 2 jcm-13-02765-f002:**
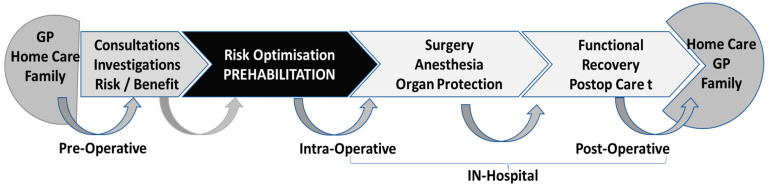
Organization plan of peri-interventional processes. GP, general physician.

**Figure 3 jcm-13-02765-f003:**
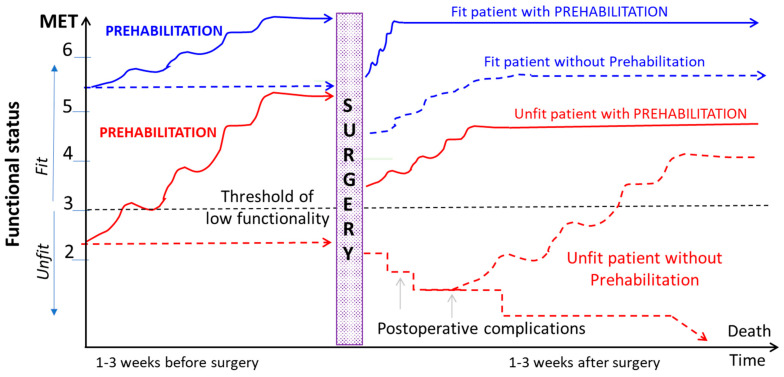
Time course of functional recovery depending on preoperative functional status assessed by MET (fit in blue, unfit in red) and on implementation of prehabilitation (solid lines, with prehabilitation; dashed line, without prehabilitation). MET, metabolic equivalent task.

**Figure 4 jcm-13-02765-f004:**
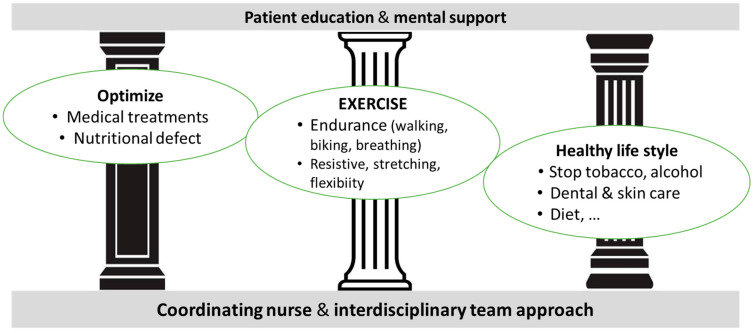
Three pillars of prehabilitation.

**Table 1 jcm-13-02765-t001:** Classification of postoperative complications.

System Involved	MILD—Self-Limited	MODERATE	SEVERE	CRITICAL
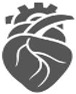 Cardiovascular	Non-sustained arrhythmiasMild hypertension or tachycardiaHypotension associated with neuraxial block	Arrhythmias requiring treatmentTransient ECG changes (ST segment depression, high T wave)Silent elevation of cTpHyper- or hypotension requiring drugs or fluidsDistal DVT	Sustained arrhythmia with unstable hemodynamicsConduction block requiring electrical pacing Symptomatic HF (stage C) Minor to moderate MI Proximal DVTMinor to moderate PE	Life-threathening arrhythmiaAdvanced HF (stage D)Extensive MI with unstable hemodynamicsExtensive PE
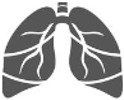 Respiratory	Inhaled O_2_ to target SpO_2_ > 92%Irritative cough, clear sputumVoice change (transient)	Hypoxemia (desaturation) requiring CPAP supportBronchospasm/laryngospasm requiring inhalation therapyPneumothorax or effusion requiring chest drainage	Hypoxemia (desaturation) requiring bronchoscopy and/or MVPneumonia; empyemaBronchopleural fistulaARDS with P/F 100–300	ARDS with P/F < 100
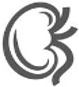 Renal	Oliguria (<0.5 mL/kg/h) for less than 6 h	KDIGO stage 1sCreat 1.5–1.9 × baseline or ≥0.3 mg/dL increaseUO < 0.5 mL/kg/h for 6–12 h	KDIGO stage 2sCreat 2–2.9 times baselineUO < 0.5 mL/kg/h for >12 hKDIGO stage 3sCreat > 3 × baseline or ≥4 mg/dL, or need for RRTUO < 0.3 mL/kg for ≥24 h	
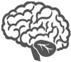 Cerebral	Drowsiness	ConfusionTransient ischemic attack	DeliriumStroke	Coma (prolonged)
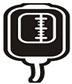 Bleeding		Minor bleeding requiring blood transfusion (<2 RBC units)	Moderate bleeding requiring 2–9 RBC units	Massive bleeding requiring > 10 RBC units or with shock
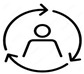 Patient-centered outcome	Incisional pain < 3/10Self-limited nausea/vomitingFatigue at exerciseMild discomfort	Incisional pain poorly responsive to analgesicsRecurrent nausea/vomitingOrthostatic hypotension	Cognitive and memory lossPartial loss of autonomy	Total loss of autonomy
Others	FeverSuperficial incisional SSI	SepsisDeep incisional SSI	Severe sepsisOrgan or space SSI	Septic shock

ARDS, acute respiratory distress syndrome; CPAP, continuous positive airway pressure; sCreat, serum creatinine; DVT, deep vein thrombosis; HF, heart failure; MI, myocardial infarct; MV, mechanical ventilation; PE, pulmonary embolism; P/F, ratio of arterial oxygen tension to fraction of inspired oxygen; RBC, red blood cell; RRT, renal replacement therapy; SSI, surgical site infection; cTp, cardiac troponin; UO, urine output.

**Table 2 jcm-13-02765-t002:** Preoperative clinical assessment and screening tools.

Risk Assessment	Score or Test	Characteristics and Interpretation
General	ASA-PS American Society of Anesthesiology-Physical status 6 classes	I: normal healthy II, mild systemic disease III, severe systemic disease, with compensated status IV, severe systemic disease that is a constant threat to life V, threat to life, moribund who is not expected to survive without treatment VI, declared brain-dead patient whose organs are being removed for donor purposes
Cardiac	Revised Cardiac Risk Index 6 items	Coronary artery disease (non-revascularized) Heart failure Renal insufficiency (serum creatinine > 2 mg/dL) Cerebrovascular disease Diabetes mellitus requiring insulin Major surgery (e.g., pneumonectomy, esophagectomy)
Pulmonary	ARISCAT Assess Respiratory Risk in Surgical Patients in Catalonia 7 items	Age (<60, 51–80, 80) Preop SpO_2_ (≥96, 91–95, ≤90%) Respiratory infection <1 month (yes/no) Preop Hb ≤ 10 g/dL Surgical incision site (peripheral, upper abdominal, intra-thoracic Duration of surgery (<2 h, 2–3 h, >3 h) Emergency procedure (yes/no)
Exercise tolerance	MET Metabolic Equivalent Task	Light intensity: <3 MET (40–55% HRMax, 20–40% VO_2_Max), writing, desk work (1.8 MET), walking 4.0 km/h (2.5 MET) Moderate intensity: 3–6 MET (55–75% HRMax, 40–60% VO_2_Max), climbing 3–4 flights of stairs or bicycling 50–100 watts (3–5.5 MET) Vigorous intensity 6–9 MET (70–90% HRMax, >60% VO_2_Max), running, 8.0 km/h (8.1 MET), rope jumping (10 MET) High intensity >9 MET (>90% HRMax, >85% VO_2_Max)
	Cardiopulmonary exercise test peak oxygen consumption (VO_2_) Stair Climbing	Low risk: peakVO_2_ > 20 mL/kg/min; > 6 floors climbing (>22 m ascension), Moderate risk: peakVO_2_ 15–20 mL/kg/min; 3–5 floors climbing (8–20 m ascension), High risk: peakVO_2_ 10–15 mL/kg/min; 1–2 floors (3–7 m ascension) Very high risk: peakVO_2_ < 10 mL/kg/min; <1 floor climbing (<2.4 m elevation)
Sarcopenia	Mini-nutritional assessment 12–14: normal 8–11: at risk <8: malnutrition	Loss of appetite (0 = severe, 1 = mild, no = 2) Loss of weight over last 3 months (0 = > 3 kg, 1 = don’t know, 3 = no loss) Motricity (0 = bed/chair, 1 = autonomous at home, 2 = can go outside) Acute illness or psychological stress over last 3 months (0 = yes, 2 = no) Neuropsychological problem (0 = dementia or depression, 1 = mild dementia, 2 = no) Body mass index (0 = 19, 1 = 19–< 21, 2 = 21– < 23, 3 => or = 23)
Frailty	Clinical Frailty Scale Scale with 9 grades	Very fit: Robust, energetic, regular exercise Well: No active disease, occasional exercise Medical problem well controlled, routine walking Vulnerable: Symptoms limit activities (slowing) Mildly frail: Need help in high order instrumental activities of daily living (finance, transportation, medications) Moderately frail: Need help for outside activities, keeping house, bathing Severely frail: Completely dependent for personal care (physical, cognitive) Severely frail: Dependent, approaching end of life (could not recover from minor illness) Terminally ill: Life expectancy < 6 month
Health Quality of life	EQ-5D Rate 5 domains (5 grades: no problem, slight, moderate, severe, extreme)	Mobility Self-care Usual activities Pain and discomfort Anxiety and depression

EQ-5D, EuroQuol five dimensions; HRMax, maximal heart rate; MET, Metabolic Equivalent Task; SpO_2_, pulsed oxygen saturation; VO_2_Max, maximal oxygen consumption.

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
