# Peer review of "Multi-Modal Prehabilitation in Thoracic Surgery: From Basic Concepts to Practical Modalities"

_jcm, 2024, doi:10.3390/jcm13102765_

Round 1

Reviewer 1 Report

Comments and Suggestions for Authors

In this review the Authors provide a thorough discussion about the role and methodology of prehabilitation in thoracic surgery. This topic has been addressed only in a limited number of previous publications, and therefore the mansucript could be of interest. The Authors could address in further detail the influence of prehabilitation on pulmonary function. 

Author Response

In this review the Authors provide a thorough discussion about the role and methodology of prehabilitation in thoracic surgery. This topic has been addressed only in a limited number of previous publications, and therefore the manuscript could be of interest. The Authors could address in further detail the influence of prehabilitation on pulmonary function. 

Thank you, we appreciate your comments and are trying to improve the paper’s content according to your proposals. More details have been provided in some parts and 11 references have been added. Of note, 6 references come from the Journal of Clinical Medicine.

As suggested, we have provided further details on the influence of prehabilitation on pulmonary function.

  • In the paragraph (5. Implementation of prehabilitation/ 5.3. Exercise training/ Type of exercise), we have reorganized and added:

P3 and 4, lines 352-355: “Interestingly, respiratory muscle training (RMT) using volume incentive spirometry or resistive threshold loading devices has been shown effective in boosting physical performances in healthy subjects83 and improving daily life autonomy of patients  with chronic disablities.84

  • In the paragraph (5. Implementation of prehabilitation/ 5.3. Exercise training/Mechanisms of training-induced improvements), we have reorganized and added:

P4, lines 373-82: The ET-induced increase in VO2max involves partial reversal of endothelial dysfunction with higher capillary density, expansion of the blood volume, increased adrenergic receptor responsiveness, improved ventricular relaxation, restoration of insulin sensitivity and enhanced oxidative mitochondrial performances in skeletal muscles owing to upregulation of PPAR and key enzymes in the tricarboxylic acid cycle.92 Accordingly, the enhanced cardiac output facilitates tissue oxygen diffusion and the mitochondrial biogenic changes lead to better utilization of oxygen within the working muscles. Both in RMT and ET, the higher ventilatory loads result in structural and functional adaptive changes within respiratory muscles, conferring greater strength and resistance to fatiguing contraction while reducing the metaboreflex.93

  • In the paragraph (5. Implementation of prehabilitation/ 5.3. Exercise training/Clinical medicine), we have reorganized and added:

P4, lines 393-6:

96,97 In a meta‑analysis of seven trials including 248 older individuals, inspiratory muscle performance was significantly improved after IMT at moderate intensity levels (30‑80% of maximal inspiratory pressure) over at least 4 weeks compared with sham treatment.98

  • In the paragraph (5. Implementation of prehabilitation/ 5.3. Exercise training/Physical training), we have reorganized and added:

P5, lines 412-4: . Most patients are capable of increasing their aerobic fitness by 1.6 to 2mL/kg/min and maximal inspiratory pressure by an average of 15 cm of water (+18%) which are considered clinically and functionally relevant changes.105,106

P5, lines 422-31: These ET-induced protective mechanisms involve cardiovascular and muscular adaptive changes (e.g., enhanced oxygen tissue delivery and extraction), allowing the postoperative patient to sustain higher ventilatory loads and to prevent alveolar collapse while facilitating clearance of bronchial secretions.

In the subset of patients undergoing lung cancer resection, although preoperative exercise training failed to achieve significant decrease in short-term mortality (RR=0.66, 95% CI 0.22 to 2.22), the overall incidence of major complications based on Dindo-Clavien score > 2 was reduced (RR=0.42, 95%CI 0.25to 0.69) while indicators of quality of life tended to improve. 

Reviewer 2 Report

Comments and Suggestions for Authors

The format of some references cited throughout the text needs to be revised. There are also some passages in the text where the concordances need to be revised to make the text even clearer and easier to understand. I would like to conclude by stating the importance of the work and the relevance of the topic in the area of post-operative rehabilitation.

Author Response

The format of some references cited throughout the text needs to be revised. There are also some passages in the text where the concordances need to be revised to make the text even clearer and easier to understand. I would like to conclude by stating the importance of the work and the relevance of the topic in the area of post-operative rehabilitation.

Thank you, we appreciate your comments and are trying to improve the paper’s content according to your proposals. More details have been provided in some parts and 11 references have been added. Of note, 6 references come from the Journal of Clinical Medicine. The modified parts of the manuscript are highlighted in yellow.

1) The paper has been partially rewritten and re-organized to convey relevant messages that would help the clinicians to select best practices throughout the perioperative journey. To facilitate reading, some sections have been further divided with sub-sections  wih subheadings :

  1. Surgical stress and physiological responses

   2.1. Neuroendocrine and inflammatory pathways

   2.2. Biological and clinical expression of the stress response

  1. Preoperative assessment and risk factors of postoperative complications

    4.1. Clinical assessment

    4.2. Cardiac and respiratory assessment

    4.3. Functional assessment

2) in the Conclusions section, the importance of postoperative rehabilitation has been emphasized

Reviewer 3 Report

Comments and Suggestions for Authors

General comment: 

The article presents a comprehensive and detailed exploration of critical aspects of thoracic surgery, including the physiological responses to surgical stress, postoperative complications, preoperative functional assessment, risk factors, and the implementation of prehabilitation programs. With some refinement for clarity, consistency, and practical applicability, it has the potential to be an excellent resource for clinicians and researchers alike.

In addition, The layout of the article needs organization and improvement; there are excessive spaces, occasionally sentences without periods, among other issues. This needs to be better presented and revised. References must adhere to the journal's format and be consistent throughout the text and in the bibliography section. The sections on Author Contributions and Funding should be completed, or deleted where not applicable.

Please consider my specific comments for each section.

Comments on the Quality of English Language

Moderate editing of English language required

Author Response

The article presents a comprehensive and detailed exploration of critical aspects of thoracic surgery, including the physiological responses to surgical stress, postoperative complications, preoperative functional assessment, risk factors, and the implementation of prehabilitation programs. With some refinement for clarity, consistency, and practical applicability, it has the potential to be an excellent resource for clinicians and researchers alike.

In addition, The layout of the article needs organization and improvement; there are excessive spaces, occasionally sentences without periods, among other issues. This needs to be better presented and revised. References must adhere to the journal's format and be consistent throughout the text and in the bibliography section. The sections on Author Contributions and Funding should be completed or deleted where not applicable.

Thank you, we appreciate your comments and the current paper version tries to address the remarks and issues you have raised. We feel this will help to improve the paper’s content and style.

1) Clarity, consistency and practical applicability

The paper has been partially rewritten and re-organized to convey relevant messages that would help the clinicians to select best practices throughout the perioperative journey. The modified parts of the manuscript are highlighted in yellow.

Table 1 has been added to illustrate an objective scoring system of postoperative complications based on target organs and severity of fysfunction.

To facilitate reading, some sections have been further divided with sub-sections  wih subheadings :

  1. Surgical stress and physiological responses

   2.1. Neuroendocrine and inflammatory pathways

   2.2. Biological and clinical expression of the stress response

  1. Preoperative assessment and risk factors of postoperative complications

    4.1. Clinical assessment

    4.2. Cardiac and respiratory assessment

    4.3. Functional assessment

2) Layout, phrasing: The full manuscript has been passed through grammar, style and spell electronic checker followed by manual review and final corrections.

3) References: As suggested in the JCM MDI author instructions, we have used Endnotes software package that does not include the MDPI format. We will see with the Editorial office how to solve this issue.

  • Sections on Author Contributions and Funding: this section is completed

Reviewer 4 Report

Comments and Suggestions for Authors

In this manuscript authors presents prehabilitation concept for patients undergoing thoracis surgery. The manuscrupt is well written and easy to follow and it presents a valuable contribution to the filed.

1. in rows 62-64, author says: "term “rehabilitation” refers to similar interventions conducted in patients with chronic diseases (e.g., chronic obstructive pulmonary disease, heart failure, stroke, diabetes mellitus) as well as after surgery".  Please rephrase this statement. It could be interpreted as rehabilitation is a term used for prehabilitation of patients with chronic disease, while it is more accurate to use "rehabilitation" in terms of postoperative care.

2. Rows 236-253: When mentioning nutritional deficits and selective supplements, it usually takes time to correct them and it takes longer time for, e.g. vitamin D supplementation to have an effect on clinical outcomes, which could lead to delaying operative procedures. More emphasis could be on restricted effects of prehabilitation on nutritional deficits as nutritonal support could be more beneficial in postoperative period

3. Please correct the references, not the same style in introduction and in the rest of the manuscript (superscript).

4. Figure 1. D0-D6 - are those postoperative days? Please explain in figure legend. Consider placing abbreviations in figure legend also.

5. row 137-139: there is referral to table in the text, but the table is missing. Same thing with rows 236-253.

6. Table 1. ASA-PS categories III and IV have same description. Please correct

7. Figure 3. needs some work:  What are data sources for this graph and on what data are the curves based on? Please add units of measure for both x and y axis. What are definitions of fit and unfit patients, how is the limit between those categories defined?

Author Response

In this manuscript authors presents prehabilitation concept for patients undergoing thoracis surgery. The manuscrupt is well written and easy to follow and it presents a valuable contribution to the filed.

Thank you, we appreciate your comments and are trying to improve the paper’s content according to your proposals.

1) The paper has been partially rewritten and re-organized to convey relevant messages that would help the clinicians to select best practices throughout the perioperative journey. All modified sections are highlighted in yellow. To facilitate reading, some sections have been further divided with sub-sections  wih subheadings :

  1. Surgical stress and physiological responses

   2.1. Neuroendocrine and inflammatory pathways

   2.2. Biological and clinical expression of the stress response

  1. Preoperative assessment and risk factors of postoperative complications

    4.1. Clinical assessment

    4.2. Cardiac and respiratory assessment

    4.3. Functional assessment

  1. in rows 62-64, author says: "term “rehabilitation” refers to similar interventions conducted in patients with chronic diseases (e.g., chronic obstructive pulmonary disease, heart failure, stroke, diabetes mellitus) as well as after surgery".  Please rephrase this statement. It could be interpreted as rehabilitation is a term used for prehabilitation of patients with chronic disease, while it is more accurate to use "rehabilitation" in terms of postoperative care.

This paragraph has been rewritten and the contention is supported by proper references.

In contrast to prehabilitation, the term “rehabilitation” refers to similar interventions conducted in two different populations of patients, those presenting with chronic debilitating diseases (e.g., chronic obstructive pulmonary disease, heart failure, stroke, diabetes mellitus) and those recovering from surgery and presenting with functional deficits.10-12

  1. Rows 236-253: When mentioning nutritional deficits and selective supplements, it usually takes time to correct them and it takes longer time for, e.g. vitamin D supplementation to have an effect on clinical outcomes, which could lead to delaying operative procedures. More emphasis could be on restricted effects of prehabilitation on nutritional deficits as nutritonal support could be more beneficial in postoperative period

This paragraph has now been rephrased as following:

Undernourished patients may benefit from personalized diets over 4 to 12 weeks to replenish muscle mass while restoring muscular strength and aerobic fitness.63 Dietary adjustments are preferentially made by prescribing the intake of high energy nutrients (~30‑40 kcal/kg/day, carbohydrates, omega‑3 fatty acids), high‑quality source of proteins (~1.5‑2 g/kg/day of protein; creatine monohydrate, essential aminoacids)  and selective supplements of vitamins and trace elements (e.g., vitamin D, folic acid, cyanocobalamine, iron).64 Provision of these multi‑ingredient mixtures in the elderly has demonstrated favorable effects on lean body mass and muscular strength, with further gains when nutrition support is combined with resistance and aerobic exercise training.65 Postoperatively, attention should be paid to resume enriched feeding in these frail patients, preferentially orally or if not possible parenterally.”

  1. Please correct the references, not the same style in introduction and in the rest of the manuscript (superscript). Corrections are done

  1. Figure 1. D0-D6 - are those postoperative days? Please explain in figure legend. Consider placing abbreviations in figure legend also.

The legend is corrected as: “Figure 1. Time course of neuroendocrine and inflammatory response to surgery.

IL, interleukin; TGF-b, transforming growth factor-b. Th2 and Th1, lymphocytes T helper 1 and 2; POD1,3,and x, postoperative day 1, 3 and undetermined.”

  1. row 137-139: there is referral to table in the text, but the table is missing. Same thing with rows 236-253.

Appropriate tables have been included

  1. Table 1. ASA-PS categories III and IV have same description. Please correct

Here is the corrected table with proper definitions

ASA-PS

American Society of Anesthesiology‑ Physical status

6 classes

·     I: normal healthy

·     II, mild systemic disease

·     III, severe systemic disease, with compensated status

·     IV, severe systemic disease that is a constant threat to life

·     V, threat to life, moribund who is not expected to survive without treatment

·     VI, declared brain-dead patient whose organs are being removed for donor purposes

  1. Figure 3. needs some work:  What are data sources for this graph and on what data are the curves based on? Please add units of measure for both x and y axis. What are definitions of fit and unfit patients, how is the limit between those categories defined?

Figure 3 is original, no source data.

This figure has been slighly modified with coded colors (see legend) and with clear notifications for the x axis (weeks preop and postop) and the y axis (MET). The low functionality level and the term „unfit“ are defined by MET < 3.

The title and legends are more descriptive as following:

Figure 3. Time course of functional recovery depending on preoperative functional status assessed by MET (fit in blue, unfit in red) and on implementation of prehabilitation (solid lines, with prehabilitation;, dashed line, without prehabilitation).

MET, metabolic equivalent task.
